# OpenForecast: The First Open-Source Operational Runoff Forecasting System in Russia

**Georgy Ayzel** [1,2,*] , **Natalia Varentsova** [3] , **Oxana Erina** [4] , **Dmitriy Sokolov** [4] ,
**Liubov Kurochkina** [5] **and Vsevolod Moreydo** [2]

1   Institute for Environmental Sciences and Geography, University of Potsdam, Potsdam 14476, Germany
2   Water Problems Institute of Russian Academy of Sciences, Moscow 119333, Russia
3   Central Administration for Hydrometeorology and Ecology Monitoring (FSBI CAHEM),
    Moscow 123995, Russia
4   Department of Hydrology, Lomonosov Moscow State University, Moscow 119991, Russia
5   State Hydrological Institute, Saint Petersburg 199004, Russia
*   Correspondence: ayzel@uni-potsdam.de

**Abstract:** The development and deployment of new operational runoff forecasting systems are a strong focus of the scientific community due to the crucial importance of reliable and timely runoff predictions for early warnings of floods and flashfloods for local businesses and communities. OpenForecast, the first operational runoff forecasting system in Russia, open for public use, is presented in this study. We developed OpenForecast based only on open-source software and data—GR4J hydrological model, ERA-Interim meteorological reanalysis, and ICON deterministic short-range meteorological forecasts. Daily forecasts were generated for two basins in the European part of Russia. Simulation results showed a limited efficiency in reproducing the spring flood of 2019. Although the simulations managed to capture the timing of flood peaks, they failed in estimating flood volume. However, further implementation of the parsimonious data assimilation technique significantly alleviates simulation errors. The revealed limitations of the proposed operational runoff forecasting system provided a foundation to outline its further development and improvement.

**Keywords:** OpenForecast; open; operational service; runoff; forecasting; Russia

## 1. Introduction

Flood-related hazards are among the most devastating weather-driven natural disasters which affect the population in vulnerable areas and cause high economic losses throughout the world [1–3]. A steadily rising world population alongside an increase in natural disasters highlights the importance of developing early-warning weather systems [4,5]. Such systems are not limited to providing only timely and reliable runoff forecasts to inform local communities about possible flooding, but can also be used by local authorities and businesses as proxies in water resource management and planning, water quality prediction, and economic loss mitigation. Therefore, the development of efficient runoff forecast communication strategies is of the same importance as robust runoff prediction methodology.

The vital importance of timely forecasts for many parties creates a high interest in the development of operational flood forecasting services worldwide. There are many established operational runoff forecasting systems in place regionally, nationally, and globally, which provide skillful forecasts up to two weeks in advance (e.g., see review papers [6–8]). However, several reasons limit the efficiency and effectiveness of current systems. First, while recent advances in numerical weather prediction (NWP) allow us to get accurate weather forecasts up to a few days in advance [9], there are still many associated uncertainties. Second, our understanding of runoff formation processes is incomplete; thus, it reflects in

additional epistemic uncertainty, which is inevitable in rainfall-runoff models [10]. Third, there is a lack of observational runoff data, which is a crucial element for successful rainfall-runoff model parameters calibration and regionalization [11], especially for remote and uninhabited regions, or countries where observational data is not always readily available. However, despite the existing problems in data availability and natural limitations in water balance components prediction, current forecasting services are valuable to their target audience [12].

Most operational runoff forecasting systems are based on a conventional technological stack; this stack includes a runoff formation model and short-range meteorological forecast data (deterministic or ensemble), which drives this model [5,7,8]. Unfortunately, all or some of these components may be subject to restricted use, which limits the ability of a system to be reproduced or reimplemented in the regions not covered by existing services. Due to this, when developing new systems or improving existing services for runoff forecasting, more attention needs to be paid to ensuring that every system's component remains open and freely accessible to the interested community. This ensures the reproduction of secure results can be guaranteed [13,14] and support for the steady development of the system by those in the community can be made [15].

While the unanimous consensus is that effective communication of produced forecasts is essential [16], there are still many open questions and challenges as to how to organize this efficiently and affordably [17]. Most of the existing services share a standard way of disseminating forecasts, through interactive websites, which provide detailed information, e.g., simulated hydrographs, flood probabilities, weather forecasts, generalized outlooks [5,8]. As social networks have become an important daily source of information, the circulation of forecasts on networks such as Facebook and Twitter has gained popularity internationally [18,19]. Although the main focus is still to provide timely and reliable predictions of river runoff, particular attention should be paid to protecting users from misinformation and misinterpretation of forecasting data [20].

While during the last few decades Russia has faced an increase in the frequency of flood-related hazards [21], the circulation of operational runoff forecasting has remained limited. Forecasts are usually issued as news by the Central Administration of the Russian Federal Service for Hydrometeorology and Environmental Monitoring (Roshydromet) or its regional branches on official websites in textual form, mainly before the spring floods (in the middle of March) or during extreme events (e.g., for 2019 spring flood http://www.meteorf.ru/press/releases/18773/, in Russian). Thus, runoff forecasts in Russia are scattered, mostly qualitative than quantitative, and not updated online. Some Russian cities that regularly suffer from spring floods spontaneously organize the monitoring of water levels in surrounding rivers using web-cameras and then develop websites to share ongoing water level data (e.g., for Tom and Kondoma rivers near Novokuznetsk http://uznt42.ru/index.php?do=static&page=vsekamery, in Russian). However, these community-driven efforts are sporadic and provide only qualitative forecasts which are limited in their value to provide early flood warnings.

In this study, we developed Russia's first open and operational online runoff forecasting system, OpenForecast (https://hydrogo.github.io/openforecast/). The present study aims to comprehensively evaluate and demonstrate the potential of OpenForecast, as well as documenting its limitations, as the operational service for short-term runoff forecasting. The primary aim of this study is to establish the interim service that would serve as a forerunner, providing a guideline for further development. The specific value of this study lies in demonstrating the utility of open data and software for making runoff forecasting workflow freely available and reproducible.

## 2. Materials and Methods

### 2.1. Study Area and the Choice of Pilot Basins

We selected two basins located in the European part of Russia as pilot study areas to evaluate the performance of OpenForecast as an operational runoff forecasting system (Figure 1): (1) Moskva R. at

Barsuki village (755 km$^2$) and (2) Seraya R. at Novinki village (293 km$^2$). Thus, Moskva R. at Barsuki is a part of the continuous monitoring system that the Department of Hydrology of the Lomonosov Moscow State University (MSU) runs at the Mozhaysk Reservoir. This monitoring is of high importance and priority regarding the fact that the Mozhaysk Reservoir is a part of the system which supplies freshwater to Moscow megapolis area. The other pilot basin, Seraya R. at Novinki, has been selected as a basin of high interest of the Central Administration for Hydrometeorology and Ecology Monitoring (CAHEM) because spring flood there and related inundation regularly cause private property and households damages. This way, timely and reliable runoff forecasts for these two introduced pilot research basins are of crucial importance for involved parties, and have a high potential for use in water quantity and quality management strategies and mitigating the risk of floods in urban areas.

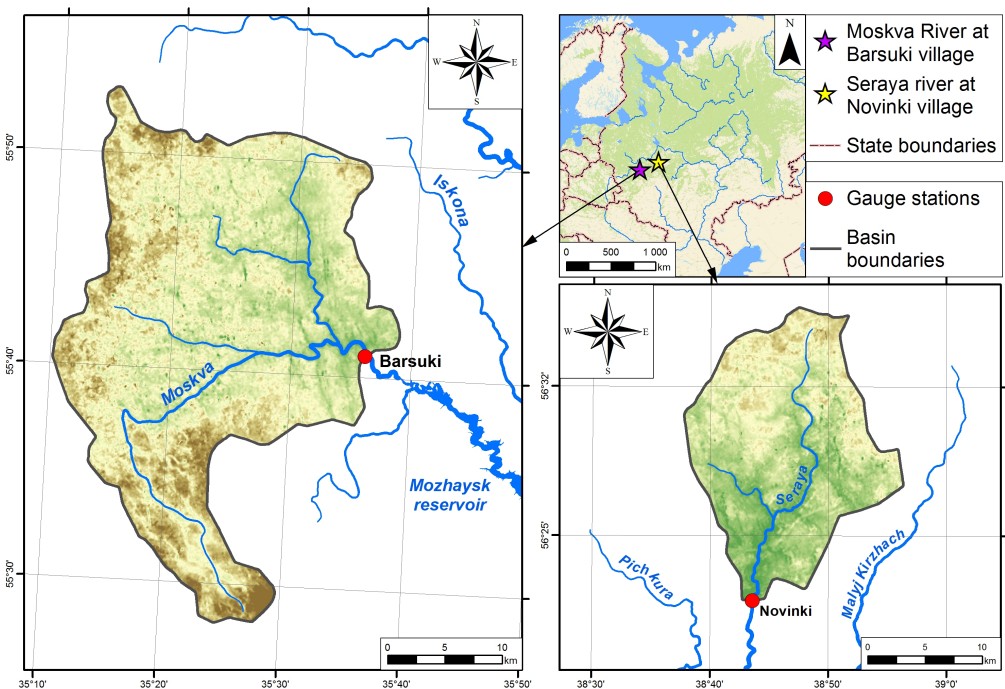

**Figure 1.** Study area and pilot basins.

*2.2. Meteorological Data*

To allow further extension of OpenForecast to a national or global scale, we use reanalysis data as a ground-truth meteorological forcing for driving hydrological model on a historical period. In the present study, we use ERA-Interim dataset by the European Centre for Medium-Range Weather Forecasts (ECMWF). ERA-Interim has a global spatial extent with a native spatial resolution of 0.75°, and a temporal resolution of 3 h. The ERA-Interim dataset covers the period from January 1979 to date [22]. The data is freely available and can be downloaded using ECMWF Web User Interface for ERA-Interim (https://apps.ecmwf.int/datasets/data/interim-full-daily/levtype=sfc/). We use 3-hourly temperature and precipitation data from the ERA-Interim archive to derive their averaged daily estimates for the period from 1 January 1979 to 30 April 2018. Era-Interim dataset showed high reliability to be used as input data for hydrological modeling in different large-scale studies [23,24].

To run OpenForecast operationally, we use data of deterministic meteorological forecasts produced by ICON—the global NWP model of the German Weather Service (DWD). ICON produces meteorological forecasts for many variables, including air temperature and precipitation; forecast runs are performed four times per day. ICON dataset has a global spatial extent with a native spatial resolution of about 13 km and a temporal resolution of 1 h [25]. The data is freely available and can be downloaded on the DWD Open Data Portal website (https://opendata.dwd.de/weather/nwp/icon/). In the present study, we use temperature and precipitation forecasts for three days ahead, while ICON

dataset provides forecast data up to one week. We also use 1-hourly estimates of temperature and precipitation data from ICON dataset which we then convert to averaged daily estimates for the period from 20 July 2018 to date. There is no study, to our knowledge, that used ICON dataset for operational runoff forecasting. As potential evapotranspiration is another required forcing variable; it was derived based on the temperature-based equation proposed by Oudin et al. [26].

### 2.3. Runoff and Water Level Observations

MSU and CAHEM provided historical runoff observations. Runoff observations cover the period from January 1979 to December 1991, and from January 2003 to December 2015 for Moskva R. at Barsuki and Seraya R. at Novinki, respectively. Because there is no systematic monitoring of discharge at pilot gauges, for OpenForecast performance evaluation, CAHEM provided the data of water levels and corresponding rating curves (Figure 2) to convert water levels to discharge estimates for the period from 20 July 2018 to 15 May 2019 at the end of May 2019.

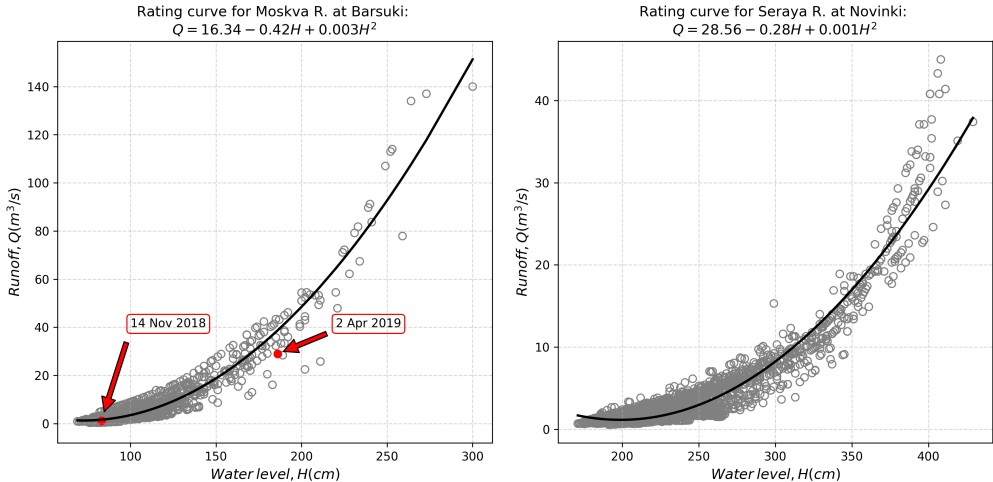

**Figure 2.** Rating curves for Moskva R. at Barsuki (**left plot**) and Seraya R. at Novinki (**right plot**). Estimated discharge and corresponding water level data for Moskva R. at Barsuki, which has been received during the two field measurement campaigns are highlighted in red dots.

Rating curves have been developed using the internal CAHEM database of observed runoff and water levels, as well as data of field campaigns for the period from 1991 to 1996 for Moskva R. at Barsuki and the period from 1991 to 2015 for Seraya R. at Novinki. A second-degree polynomial has been used to fit water level to runoff observations. High determination coefficients ($R^2$) between the observed runoff and the runoff calculated as a function of water level show high reliability of the developed rating curves ($R^2$ is 0.94 for Moskva R. at Barsuki, and 0.93 for Seraya R. at Novinki). Additionally, the comparison of discharges and water levels which have been measured during the two field campaigns on Moskva R. at Barsuki with corresponding rating curve (Figure 2, left plot) confirms high reliability of the provided rating curve.

### 2.4. Hydrological Model

In the present study, we use GR4J—the conceptual hydrological model which represents a wide range of runoff formation processes at a basin scale [27]. To represent snow formation and transformation processes, we coupled GR4J model with the Cema-Neige snow routine introduced by Valéry et al. [28,29]. GR4J showed a good performance in runoff predictions for many basins in different geographical conditions worldwide [30,31], as well as it showed a good potential for runoff predictions using meteorological reanalysis data as input [32,33].

GR4J has a lumped structure with six parameters, two of which are related to the Cema-Neige snow model (Figure 3, Table 1). To produce runoff simulations at a daily temporal resolution, GR4J

also requires daily time series of air temperature, precipitation, and potential evaporation as inputs. GR4J is distributed as part of open-source software package airGR [34].

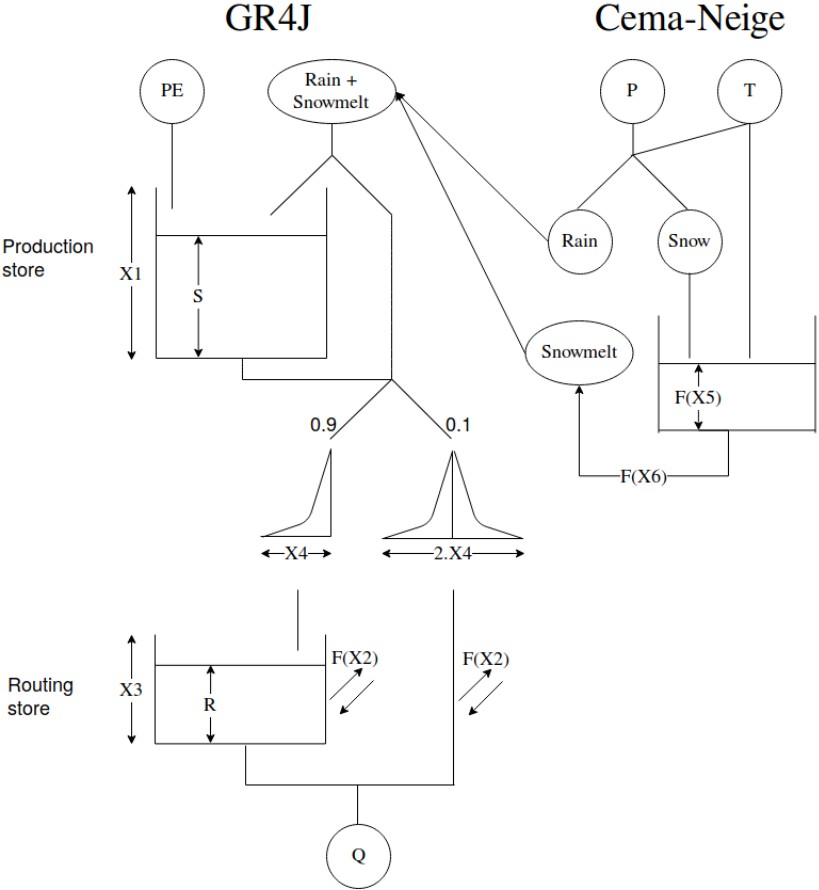

**Figure 3.** Conceptual schemes of GR4J hydrological model [27] and Cema-Neige snow accounting routine [28,29].

**Table 1.** Description and calibration ranges for GR4J and Cema-Neige free parameters.

| Parameters | Description | Calibration Range |
|---|---|---|
| **GR4J** | | |
| X1 | Production store capacity (mm) | 0–3000 |
| X2 | Intercatchment exchange coefficient (mm/day) | −10–10 |
| X3 | Routing store capacity (mm) | 0–1000 |
| X4 | Time constant of unit hydrograph (day) | 0–20 |
| **Cema-Neige** | | |
| X5 | Dimensionless weighting coefficient of the snowpack thermal state | 0–1 |
| X6 | Day-degree rate of melting (mm/(day*C°)) | 0–10 |

We use the global optimization algorithm of differential evolution [35] to find GR4J optimal parameters which led to the maximum of the Nash and Sutcliffe [36] efficiency (NSE, Equation (1)). The calibration range of the model parameters (Table 1) has been set according to studies [29,37].

We use the standard split-sample test proposed by Klemeš [38] to calibrate and evaluate the performance of GR4J model on different periods. Thus, for Moskva R. at Barsuki we perform a split-sample test on two periods: (1) from 1979 to 1985, and (2) from 1986 to 1991; and for Seraya R. at Novinki: (1) from 2003 to 2009 and (2) from 2010 to 2015, respectively. Since the choice of calibration and validation periods is subjective and may lead to significant differences in performance evaluation

results [31,39], our splitting methodology was guided only by the approximate equality of the duration of these periods.

According to Arsenault at al. [31] which identified that calibration on an entire period of observed runoff data always leads to better results on independent evaluation period, we also calibrate GR4J model against an entire runoff observations record. This way, we obtained three sets of optimal model parameters (two from the split-sample test, and one from the calibration against an entire observed runoff time series) which we further use for hydrological model performance evaluation, and runoff forecasting in ensemble manner.

To evaluate hydrological model performance regarding runoff simulations, we use two widely used criteria: the NSE, Equation (1) and the systematic error of runoff estimation (Bias, Equation (2)). This choice allows us to estimate different aspects of simulated hydrographs to better understand the efficiency of runoff formation processes representation by the hydrological model [40].

$$\text{NSE} = 1 - \frac{\sum_\Omega (Q_{sim} - Q_{obs})^2}{\sum_\Omega (Q_{obs} - \overline{Q_{obs}})^2} \tag{1}$$

$$\text{Bias} = \frac{\sum_\Omega (Q_{sim} - Q_{obs})}{\sum_\Omega (Q_{obs})} * 100\% \tag{2}$$

where $Q_{sim}$, $Q_{obs}$, $\overline{Q_{obs}}$ are simulated, observed and mean observed runoff, and $\Omega$ is the period of evaluation. According to Moriasi et al. [41], model runoff simulation can be considered to be satisfactory if NSE > 0.5, and if bias ±25%.

## 2.5. Openforecast Computational Framework

The schematic representation of the OpenForecast computational framework is shown in Figure 4. The OpenForecast runoff forecasts run operationally using standard Office PC. Forecast cycle begins daily at 06:00 UTM to produce runoff forecasts for two pilot basins for the next three days (the day of the initial run is included).

First, we download ERA-Interim air temperature and precipitation data for the period from 1 January 1979 to 30 April 2018, which is already preprocessed and stored locally. As the first OpenForecast launch was on 20 July 2018, we must fill almost three months of missing meteorological data (i.e., May, June, and until 19 July 2018, inclusively). We decided to fill them by climatological estimates—e.g., missing air temperature value for 1 June 2018 had been filled by the mean air temperature for the 1st of every June from 1979 to 2017 (following ERA-Interim time span). We consider that implemented filling method may affect the performance of GR4J hydrological model, but it is a forced compromise until new ERA-Interim data comes in to fill the missing period. In the end, we have daily time series of air temperature and precipitation for the period from 1 January 1979 to 19 July 2018 based on ERA-Interim dataset.

Second, we download ICON data from 00:00 run from the DWD Open Data Portal (https://opendata.dwd.de/weather/nwp/icon/) and preprocess it to retrieve daily averages of air temperature and precipitation. We accumulate this data locally from 20 July 2018 to date.

Third, we concatenate ERA-Interim with ICON data to obtain consistent time series of air temperature and precipitation from 1 January 1979 to the date we issue a runoff forecast. Then we additionally calculate potential evapotranspiration based on temperature-based equation proposed by Oudin et al. [26].

Fourth, we feed obtained consistent data of temperature, precipitation, and potential evapotranspiration to GR4J model which uses three sets of optimal model parameters (see Section 2.4) to produce three corresponding scenarios of continuous runoff forecast. At the last step, we calculate minimum, maximum, and average estimates of forecast runoff scenarios to update OpenForecast website with the new information.

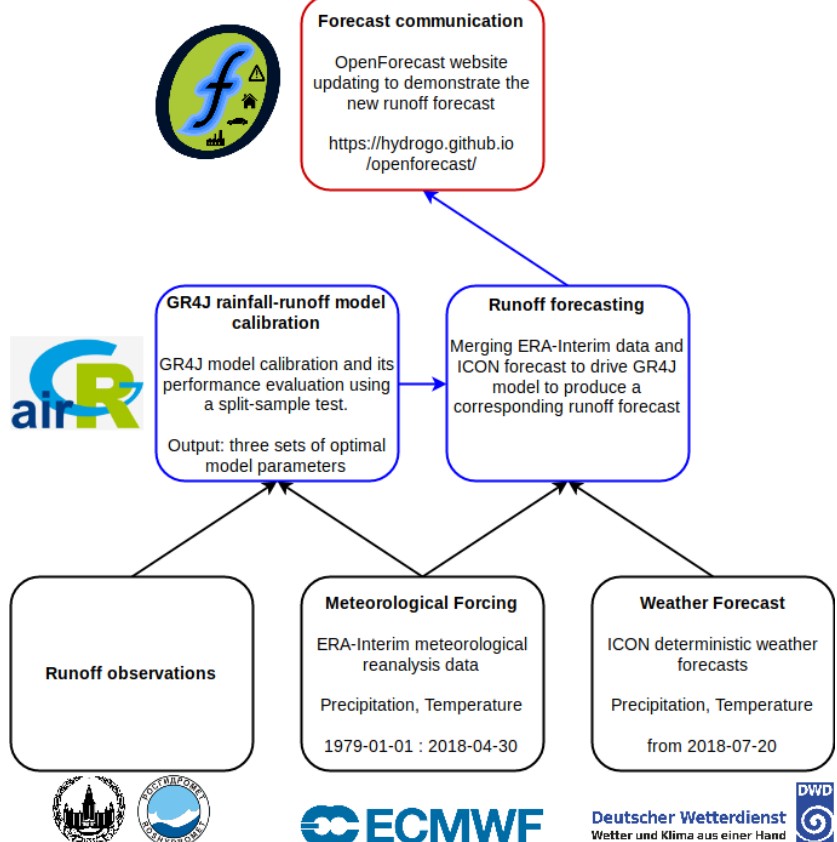

**Figure 4.** Illustration of OpenForecast computational framework.

The entire OpenForecast's workflow is automated and runs on an everyday basis without any manual intervention. As OpenForecast requires only historical runoff observations and watershed boundaries as input for the operational forecasting routine, the service can be easily expanded to basins that meet these soft requirements.

OpenForecast computational framework entirely relies on open-source software packages. We use xarray [42] to preprocess ERA-Interim and ICON data, numpy [43] and pandas [44] for general-purpose data analysis and calculation, scipy [45] for implementation of global optimization algorithm of differential evolution, and matplotlib [46] and bokeh [47] for plotting.

## 3. Results and Discussion

### 3.1. Hydrological Model Calibration and Evaluation on a Historical Period

Table 2 summarizes the comparison of the statistical model performance criteria for a runoff prediction for the calibration and validation periods, and the entire available runoff observations period for the Moskva R. at Barsuki and Seraya R. at Novinki. Additionally, Table 3 shows GR4J optimal model parameters which were calibrated for the different periods.

**Table 2.** Model performance summary for Moskva R. at Barsuki and Seraya R. at Novinki for different periods. The numerator and denominator values stand for NSE and Bias, respectively.

| **Moskva R. at Barsuki** | | | |
|---|---|---|---|
| **Periods for:** | **Validation (in Columns)** | | |
| **Calibration (in Rows)** | **1979–1985** | **1986–1991** | **1979–1991** |
| 1979–1985 | 0.77/2.8 | 0.75/−6.6 | 0.78/−1.6 |
| 1986–1991 | 0.66/12 | 0.79/2.1 | 0.75/7.5 |
| 1979–1991 | 0.76/6 | 0.78/−3.5 | 0.79/1.6 |
| **Seraya R. at Novinki** | | | |
| **Periods for:** | **Validation (in Columns)** | | |
| **Calibration (in Rows)** | **2003–2009** | **2010–2015** | **2003–2015** |
| 2003–2009 | 0.76/−10 | 0.62/−26 | 0.69/−17 |
| 2010–2015 | 0.61/16 | 0.77/−11 | 0.7/3.5 |
| 2003–2015 | 0.72/−5.2 | 0.75/−24 | 0.74/−14 |

**Table 3.** Obtained optimal model parameters for different calibration periods.

| | **Moskva R. at Barsuki** | | | **Seraya R. at Novinki** | | |
|---|---|---|---|---|---|---|
| **Parameters** | **1979–1985** | **1986–1991** | **1979–1991** | **2003–2009** | **2010–2015** | **2003–2015** |
| X1 | 110 | 113 | 81 | 240 | 262 | 384 |
| X2 | −0.1 | 0.1 | 0 | −3 | −0.8 | −2.1 |
| X3 | 48 | 33 | 37 | 65 | 51 | 55 |
| X4 | 2.3 | 2.4 | 2.4 | 2.3 | 2.3 | 2.4 |
| X5 | 0.8 | 0.8 | 0.8 | 0.9 | 0.9 | 0.9 |
| X6 | 3.9 | 2.9 | 3.4 | 2.9 | 2.1 | 2.2 |

As can be seen from Table 2, GR4J model shows robust evaluation performance for any calibrated set of model parameters. The most significant drop in model performance related to the situation when the model was calibrated against runoff record from 1986 to 1991 and then evaluated on the previous period from 1979 to 1985: NSE drops from 0.79 to 0.66, and bias increases from 2.1 to 12. The reason of this behavior is that during the period from 1979 to 1985 runoff at Moskva R. at Barsuki is characterized by more considerable intra-annual variability and more complex patterns of spring floods and followed spring flash floods, thus, model parameters calibrated on a different period (1986–1991) do not catch such a complexity which was not introduced during the calibration period. It also reflects the significant difference between X3 (routing store capacity) and X6 (snowmelt rate) parameter values, which have a strong influence on a spring flood representation. Model parameters set which was obtained from the calibration against the entire period underlines the feature of high GR4J model adaptation to different runoff scenarios.

Table 2 shows the same pattern of model robustness for Seraya R. at Novinki as has been detected for Moskva R. at Barsuki, but for both variants of calibration and validation period splits. This can be explained by the distinct differences of observed runoff records between two periods. For example, the period from 2010 to 2015 has no spring flood events for two years of 2014 and 2015 and introduces lower frequency of winter flash floods than the period from 2005 to 2009. However, as also has been shown for Moskva R. at Barsuki, GR4J model which was calibrated against the entire runoff record shows good results for both periods that approves its flexibility to be tuned to represent runoff time series which imply high interannual variability. The identified variation of optimal model parameters values confirms the high contrast between calibration periods.

Results show that X1 (production store capacity), X3 (routing store capacity), and X6 (snowmelt rate) parameters of GR4J model vary the most to adapt the model to contrasting calibration periods (Table 3). This fact, concerted with model evaluation results (Table 2) shows that GR4J model has a potential to be used for operational runoff forecasting in the case that there will be

no substantial differences between observed runoff patterns and dynamics on a historical period in comparison with the period of operational use. Obtained results confirm findings presented by Ayzel [32], who indicated the limited robustness of GR4J model for runoff predictions in three basins located in the north of Russia.

### 3.2. Operational Runoff Forecasting with OpenForecast

We launched OpenForecast on 20 July 2018. At the moment, OpenForecast has been in operation for almost a year, providing runoff forecast for two pilot basins for three days ahead.

Operational control of runoff forecast for Moskva R. at Barsuki has been carried out based on the Pearson correlation coefficient estimation between predicted runoff and observed water levels which were available on the "ESIMO" website (http://esimo.ru/dataview/viewresource?resourceId= RU_RIHMI-WDC_1325_1). Results of an operation control show that for every considered lead time estimated correlation coefficient between observed water level and runoff forecast is around 0.94. Unfortunately, there was no operational information about water levels on Seraya R. at Novinki to apply such an operational control.

The evaluation of OpenForecast operational results has been carried out at the end of May 2019, when we received observed water levels and rating curves for the pilot basins from CAHEM. Then, water level observations were converted to discharge using provided rating curves (Figure 2).

As Russia faced similar exceptional drought conditions as having been recorded for Central-Northern Europe (European Drought Observatory, http://edo.jrc.ec.europa.eu), there was not any considerable flashflood during summer or autumn periods for both pilot basins. This way, while we calculate performance statistics for the entire period of OpenForecast operational run from 20 July 2018 to 17 May 2019 (Table 4), we plot the only period from 1 March 2019 to 30 April 2019 to demonstrate the most interesting results of spring flood simulation (Figures 5 and 6).

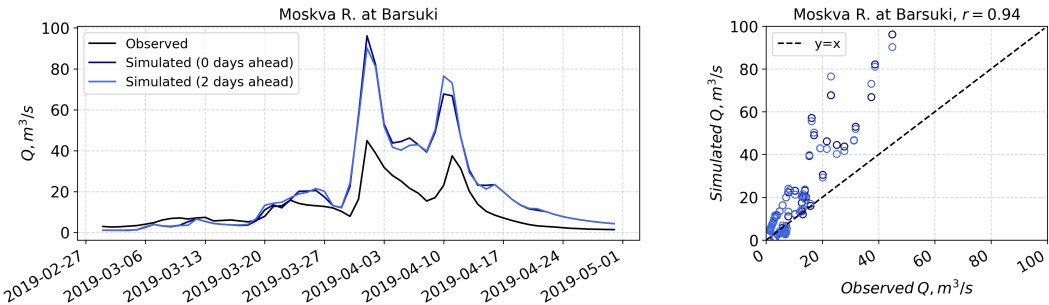

**Figure 5.** Hydrographs (**left plot**) and scatter plot (**right plot**) of observed and simulated discharges for Moskva R. at Barsuki during the 2019th spring flood.

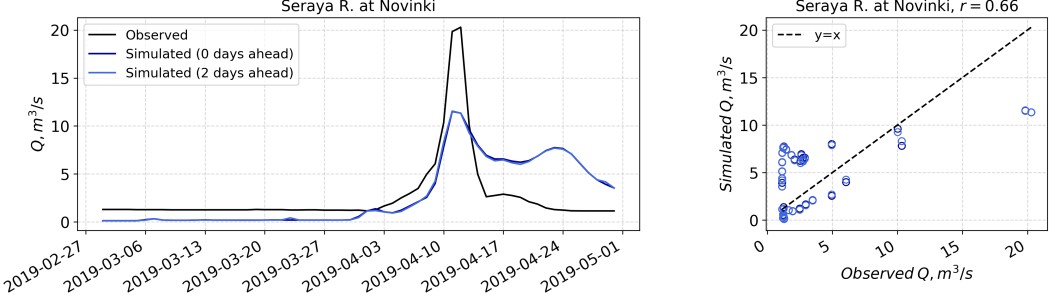

**Figure 6.** Hydrographs (**left plot**) and scatter plot (**right plot**) of observed and simulated discharges for Seraya R. at Novinki during the 2019th spring flood.

There is a clear pattern of runoff overestimation by OpenForecast for Moskva R. at Barsuki (Figure 5). Negative NSE and high bias values for Moskva R. at Barsuki show that

OpenForecast demonstrates unsatisfactory performance regarding operational runoff predictions ([41]). Although OpenForecast efficiency in terms of negative NSE and high bias shows its limited reliability, we want to emphasize its overall utility for users. The dates of the beginning of the spring flood and occurrence of two peak discharges have been accurately predicted, as well as the shape of the spring flood hydrograph. This way, the added value of OpenForecast predictions is much higher in comparison to the mean model which NSE is zero.

Figure 6 shows the results of operational forecasting for Seraya R. at Novinki. While low NSE and high bias for Seraya R. at Novinki (Table 4) indicate the unsatisfactory performance of OpenForecast, we predict the timing of the spring flood—the beginning date, and the date of the peak discharge—correctly. Opposite to results for Moskva R. where we overestimate runoff for the entire spring flood period, for Seraya R. we underestimate runoff at the beginning of the event and significantly overestimate it during the flood recession period.

**Table 4.** OpenForecast performance summary for the period of its operational evaluation. The numerator and denominator values stand for NSE and Bias, respectively.

| Basin | 0 Days Ahead | 1 Day Ahead | 2 Days Ahead |
|---|---|---|---|
| Moskva R. at Barsuki | −0.48/43 | −0.48/43 | −0.51/45 |
| Seraya R. at Novinki | 0.18/−37 | 0.18/−37 | 0.19/−38 |

As GR4J model demonstrated minor limitations regarding its robustness for runoff modeling for Moskva R. at Barsuki on a historical period, the results of operational runoff forecasting performance evaluation introduce the high systematic bias. This is contrary to the study of Unduche et al. [48], where authors showed an apparent loss of model robustness on a validation period, but the same model continues to provide good results during the operational use. We hypothesize about two possible reasons for OpenForecast low efficiency, which was demonstrated during the operational run for Moskva R. at Barsuki.

First, in contrast to Unduche et al. [48] which did a split of observed data in the manner that the period of model calibration is directly followed by the period of operational use, in our setting there is a significant gap of 17 years between the last year of the calibration period (1991) and the beginning of OpenForecast operational use (2018). Data analysis of observed runoff and precipitation data shows that 2019th spring flood has lower peak discharge and volume in comparison with floods occurred until 1992, but the amount of accumulated precipitation for the period before spring flood (from the beginning of October to the end of March) is comparable with mean estimation for that period. Thus, it is possible that due to recent climate-induced changes in spring flood dynamics [21] GR4J model lost its robustness and did not represent current runoff formation patterns, e.g., the capacity of soil water reservoir could change significantly for 17 years to reflect drying trends for soil moisture [49].

Second, parsimonious Cema-Neige snow routine does not represent complex snow cover dynamics, which can lead to significant errors in snow accumulation estimates, and as a result, to corresponding errors in spring flood volume simulation. It worth mentioning that identified high bias in runoff forecast during the operational use does not seem to have anything to do with the difference between daily and monthly precipitation estimates obtained from ERA-Interim and ICON data products (Figure 7).

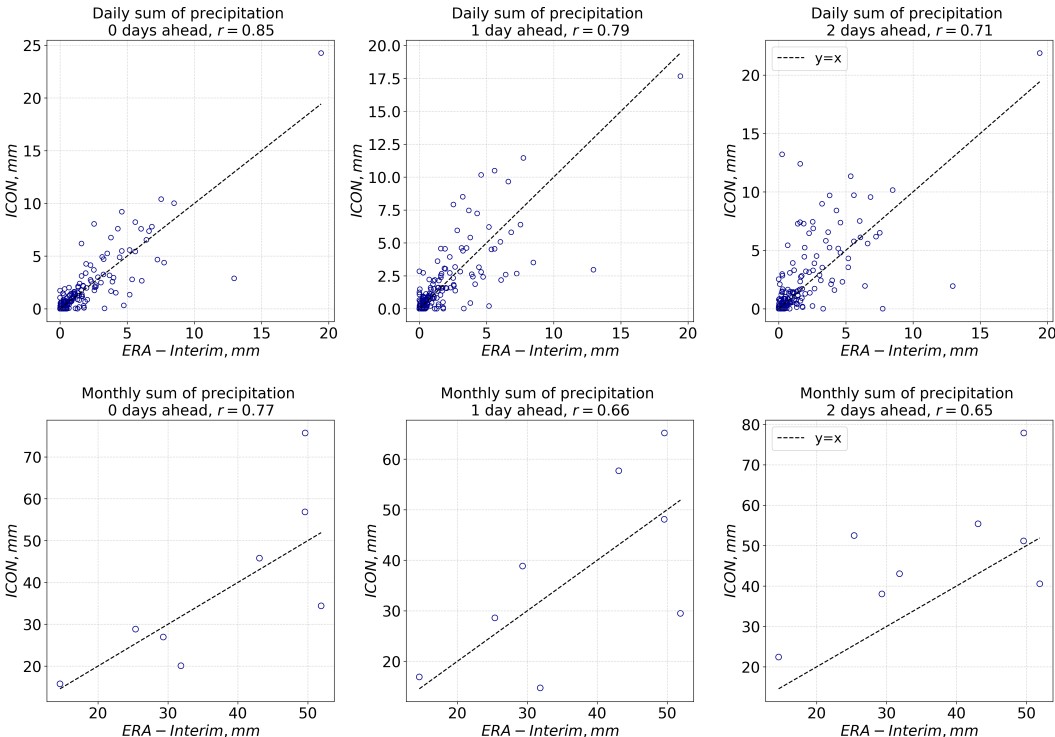

**Figure 7.** Relationship between daily (**top row**) and monthly (**bottom row**) precipitation estimates provided by ERA-Interim and ICON data for Moskva R. at Barsuki.

In contrast to Moskva R. at Barsuki where the 17-year gap between calibration and operational periods is introduced, for Seraya R. at Novinki, this gap consists only of 3 years. Thus, GR4J optimal model parameters must show a higher degree of robustness for Seraya R. However, and there were no distinct spring flood events on Seraya R. at Novinki (high runoff amplitude, long-lasting duration) for 2014 and 2015. These recent changes in spring flood characteristics concerted with exceptional drought during the summer-autumn period of 2018 may cause revealed substantial differences between observed and simulated runoff. Also, as the gauge station on Seraya R. at Novinki is non-operational, it is possible that data which has been obtained on this gauge is less reliable. The last hypothesis about the low efficiency of OpenForecast in runoff predictions goes to the fact that the basin area of Seraya R. at Novinki includes the town of Aleksandrov and a large proportion of urban territories, which may play a significant role in runoff formation and transformation processes there.

Results also identify the small difference between simulated runoff forecasts for 0 and 2 days ahead (Figures 5 and 6, Table 4). This can be explained by high consistency and skill of ICON weather forecasts, which is also confirmed by the comparison with ERA-Interim data (Figure 7).

### 3.3. Operational Use of OpenForecast in CAHEM

Independently, OpenForecast predictions were used by the CAHEM for operational runoff forecasting as one of the possible realizations. As during the OpenForecast evaluation period CAHEM had access to both operational water levels and rating curves, it introduced the parsimonious way of OpenForecast predictions updating by observational data assimilation [50] (Equation (3)), as follows:

$$Q_{upd[0,1,2]} = Q_{f[0,1,2]} * \frac{Q_{o[-1]}}{Q_{f[-1]}} \tag{3}$$

where $Q_{upd}$—updated runoff forecast, $Q_f$—OpenForecast runoff prediction, $Q_o$—observed runoff, $[-1, 0, 1, 2]$—pointers for daily time steps where 0 is the day of forecast. The proposed data assimilation technique can be considered to be the simplified version of the technique proposed by Kneis et al. [51].

Figures 8 and 9 show updated OpenForecast predictions for Moskva R. at Barsuki and Seraya R. at Novinki, respectively, using observational data assimilation technique by CAHEM in comparison with default OpenForecast predictions and observations. Results show that the introduced parsimonious data assimilation technique significantly improves default OpenForecast runoff forecast.

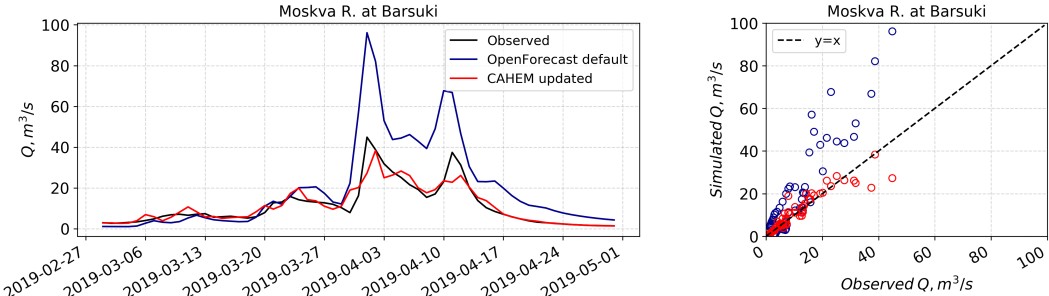

**Figure 8.** Results of parsimonious data assimilation technique proposed by CAHEM for operational runoff forecasting of the spring flood 2019 for Moskva R. at Barsuki: hydrographs (**left plot**) and scatter plot (**right plot**) of observed and simulated discharges.

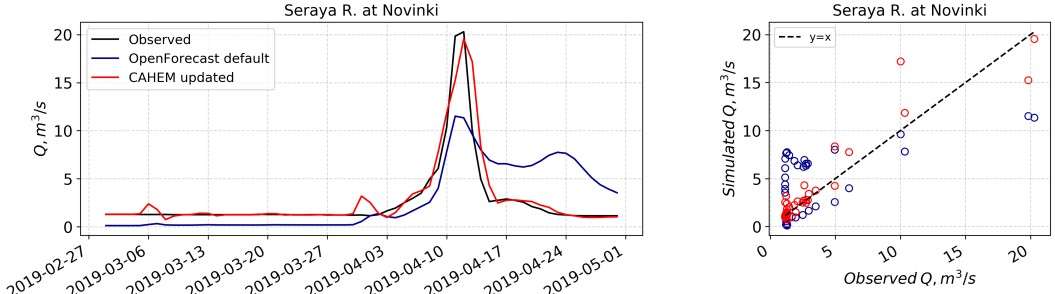

**Figure 9.** Results of parsimonious data assimilation technique proposed by CAHEM for operational runoff forecasting of the spring flood 2019 for Seraya R. at Novinki: hydrographs (**left plot**) and scatter plot (**right plot**) of observed and simulated discharges.

Table 5 summarizes the efficiency of the introduced data assimilation technique in terms of NSE and Bias performance criteria. For Moskva R. at Barsuki data assimilation turns NSE from −0.48 to 0.91 and significantly reduces bias to almost zero; for Seraya R. at Novinki NSE increases from 0.18 to 0.89 and bias drops from −37 to near 3%. Thus, data assimilation technique has transformed runoff forecasting results from unsatisfactory to very good [41]. This way, our study confirms the positive effect of observed runoff data assimilation for runoff forecasting also presented by Kneis et al. [51]. However, until Russian authorities do not provide automated access to water level observations, this information cannot be used in OpenForecast operational setup.

**Table 5.** Performance summary of updated runoff forecast using CAHEM observational data assimilation scheme for the period of OpenForecast operational evaluation. The numerator and denominator values stand for NSE and Bias, respectively.

| Basin | 0 Days Ahead | 1 Day Ahead | 2 Days Ahead |
| --- | --- | --- | --- |
| Moskva R. at Barsuki | 0.91/−0.1 | 0.91/0 | 0.90/1 |
| Seraya R. at Novinki | 0.89/2.6 | 0.89/2.6 | 0.89/2.9 |

*3.4. Communication of Forecasts*

OpenForecast has different channels to communicate its runoff forecasts to a broad audience. The main channel is OpenForecast website—https://hydrogo.github.io/openforecast/. We have built this website using GitHub Pages—an open and free of charge platform for hosting static web pages.

For every pilot basin, we generate an individual web page using bokeh plot library [47] where we show runoff predictions from the beginning of the OpenForecast operational run alongside recent runoff forecast for the next three days (Figure 10). Interactive nature of generated web pages allows the user to scroll and zoom over the figure investigating simulated hydrographs in detail. We also communicate corresponding hydrological model uncertainty by showing forecast spread among three scenarios we calculate using three different sets of optimal parameters.

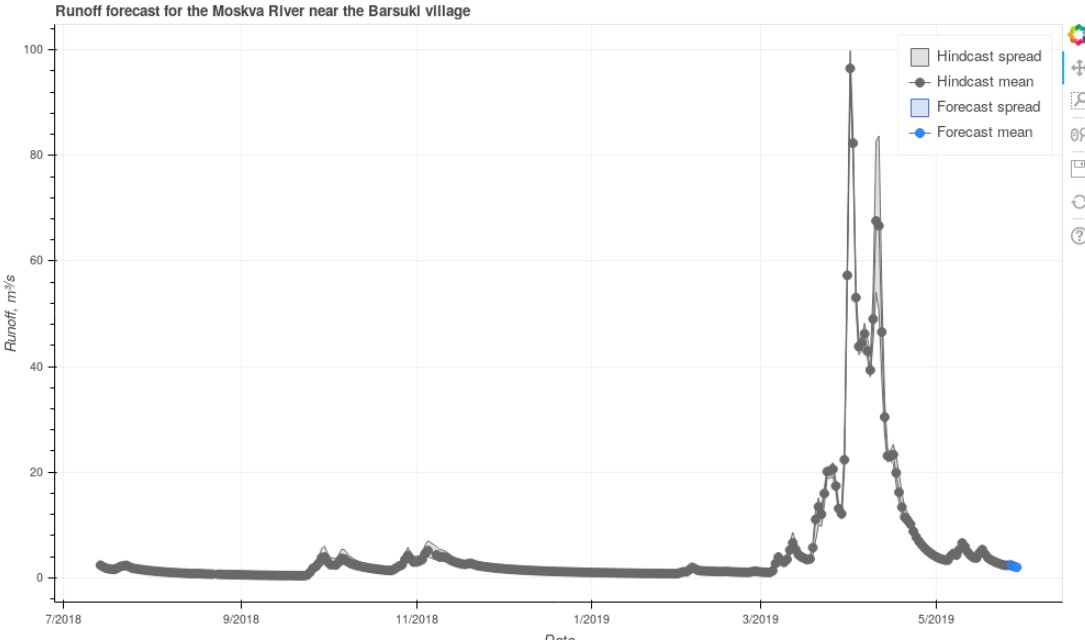

**Figure 10.** Screenshot of OpenForecast website with the runoff forecast for Moskva R. at Barsuki (https://hydrogo.github.io/openforecast/barsuki.html, 27 May 2019).

In recognition of the growing popularity of messengers and social networks, we also share OpenForecast results using the Telegram channel (https://t.me/showmethebest/) and the Instagram account of the MSU Krasnovidovo Research Station (https://www.instagram.com/krasnovidovo.msu/) (for examples see Figure 11). Trying to involve a wider audience, we have written the news report on the website of MSU Department of Hydrology (http://www.geogr.msu.ru/news/news_detail.php?ID=13421, in Russian). Unfortunately, we did not get any feedback outside our research group regarding OpenForecast operational functioning. To intensify forecasts communication, we will try to do more periodical updates and introduce some new formats, e.g., videos.

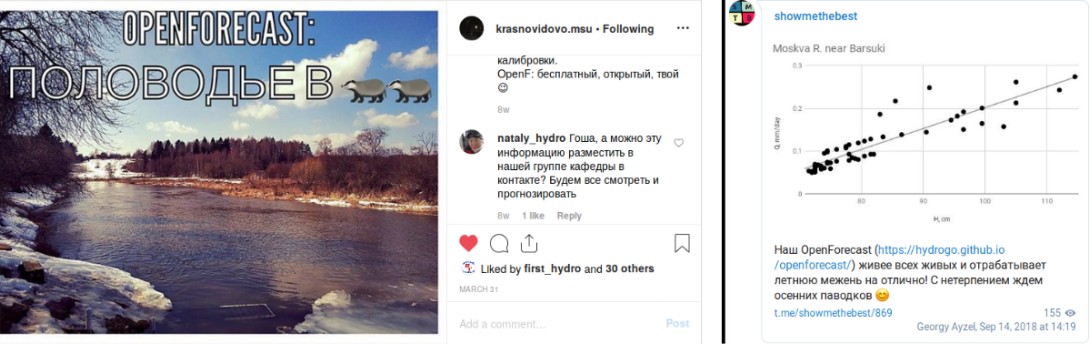

**Figure 11.** Screenshots of the posts in Instagram (**left plot**, https://www.instagram.com/p/BvrLbBXFT7C/, 31 March 2019) and Telegram (**right plot**, https://t.me/showmethebest/869, 14 September 2018) to support OpenForecast results communication in social media (in Russian).

## 4. Conclusions

In the present paper, we describe our motivation and following production of OpenForecast—the first open-source operational runoff forecasting system in Russia—which was developed for and evaluated on two pilot basins in the European part of Russia. OpenForecast uses only open-source and freely available data and software as framework's building blocks; thereby it follows principles of open and reproducible science, and moreover, has a potential to be implemented at a national scale, or even globally.

The initial evaluation results of OpenForecast on almost a year of continuous operational use are promising, showing good skill in flood timing prediction, i.e., date of the beginning of the flood period or date of the peak discharge. However, initial results also indicate the significant inconsistencies between simulated and observed flood volumes for both pilot basins. Therefore, at the present state of the OpenForecast development, produced runoff forecasts should not be used as a reliable reference. Revealing of the particular reasons which may cause reported errors in runoff prediction will be in the main focus of following studies. We believe that for the next update of OpenForecast operational setup, we must focus our attention on improving the underlying hydrological model as it showed limited robustness on the evaluation period. There are two most perspective directions for further model improvement. First, to re-calibrate the model against the data we obtained from CAHEM for the evaluation period. Second, to develop an error model for addressing discrepancies between simulated and observed runoff.

Additionally, we demonstrate the utility of OpenForecast for operational use in the CAHEM. Using OpenForecast runoff forecasts as a first-guess forecast, CAHEM introduces the parsimonious data assimilation technique which uses recently observed runoff data for a further forecast updating procedure. This data assimilation technique significantly improves runoff prediction efficiency, and additionally underlines and confirms the importance of observational data assimilation for getting reliable hydrological modeling results [50,52].

OpenForecast is running operationally and communicates runoff forecast for the next three days via the openly available website at https://hydrogo.github.io/openforecast/. OpenForecast development and evaluation results proved the concept of the proposed computational framework, but also highlighted the necessary need for further improvements. Specifically, improvements of OpenForecast are currently underway in four directions:

1. *Scientific development.* As initial evaluation results indicated some problems in flood prediction efficiency, we want to understand the possible sources of OpenForecast errors better.
2. *Software development and service maintenance.* Under this direction, we plan to migrate from ERA-Interim to ERA-5 meteorological reanalysis data (both produced by ECMWF), from deterministic (ICON) to ensemble weather forecast product (ICON-EPS) of DWD, and from the one hydrological model (GR4J) to the family of GR models (e.g., GR5J, GR6J). We also want to document and then share OpenForecast code to engage the hydrological community in further development.
3. *Communication of forecast.* We want to promote OpenForecast to a broader audience: both specialist and non-specialist should be able to benefit from our service to make informed decisions.
4. *OpenForecast expansion.* As OpenForecast requires only historical runoff observations and watershed boundaries as input for initialization of operational forecasting routine, we want to expand our service for as many basins as satisfy these conditions disregarding their location.

**Author Contributions:** G.A. developed OpenForecast, evaluated its performance, and wrote the paper. N.V., O.E., D.S. obtained and processed the runoff data, and organized field campaigns. N.V. developed CAHEM data assimilation technique. L.K. developed OpenForecast GIS. N.V., O.E., D.S., L.K., V.M. revised the paper.

**Funding:** The present study was mainly funded by the Russian Foundation for Basic Research (RFBR), project no. 19-05-00087 A. The hydrological model development and evaluation part (Section 2.4) was supported by the Russian Science Foundation, project no. 16-17-10039. Georgy Ayzel was financially supported by Geo.X, the Research Network for Geosciences in Berlin and Potsdam.

**Acknowledgments:** This paper was possible thanks to an existing partnership of young hydrologists in many academic institutions who want to drive the development of hydrological research in Russia using all the opportunities provided by modern science, despite existing barriers. The authors thank the HEPEX community (https://hepex.irstea.fr/) for its inspiring work in the field of runoff forecasting. G.A. thanks Maik Heistermann and Lisei Koehn-Reich for fruitful discussions about OpenForecast verification setup.

**Conflicts of Interest:** The authors declare no conflict of interest.

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
