# Peer review of "OpenForecast: The First Open-Source Operational Runoff Forecasting System in Russia"

_water, doi:10.3390/w11081546_

Reviewer 1 Report

The manuscript presents a runoff forecasting system in Russia for early warning of floods. While the topic is very promising given the complexity of the system due to the snowmelt-driven runoff, the manuscript lacks the scientific intensity that is expected for a publication in a reputed journal. The authors provided lots of background information about the utility of the paper in the introduction. However, they failed to depict the effectiveness of the developed forecasting system scientifically. Further, the writing style was very intricate, which makes the paper hard to follow. Some of the paragraphs and sentences read like a project report rather than a scientific manuscript. Overall, I do not think the document is ready for publication in its current form. It requires rigorous analysis and effort to describe the utility of the system rather than just providing the results. My comments about are below:

General comments:

1.    The method section is mostly inadequate. The core of the system is the hydrological model. However, there is no description of model development. What are the parameters used to calibrate the hydrological model? And what are the most sensitive one?  Scientifically, how the hydrological models are coupled with the other components? The data used to develop the system is vaguely expressed. The authors need to list the collected data set along with their temporal scale. The QA/QC of the data also needs to be addressed. 

2.    There are too many texts in the introduction and in the overall manuscript that was not well-articulated. The articulation was missing in writing – instead, the paper reads like a lengthy description of different aspects. I suggest the authors do completely rewrite the introduction section – the section needs to be summarized logically other that using wordy, vague sentences. There are some unnecessary sentences which need to be either removed or revised. The last paragraph of the introduction needs a complete revision.  Currently, it reads like an executive summary of a project report. The authors need to indicate the major objectives of the study in the last paragraph. 

3.    While lots of promises were made in the introduction about the developed system, the forecast performance was somewhat disappointing (Fig 4,5 and Table 3). The NS values were negative, which indicates that mean is a better model than the developed system. A sensitivity analysis of the parameters would be helpful for a potential improvement of the model performance. The authors argued that although quantitatively the model is under or over-predicting, it is capturing the flush flood picks and shape of the hydrograph. Here, I am interested to know what boundary conditions were used in the model upstream and downstream. If the model is fixed by a downstream water level boundary condition, there is a likelihood that the shape of the predicted runoff time-series will reflect the boundary condition. That is why, in the methodology, it is essential the clearly describe the model development process. I would also like to see a plot to compare the shapes of the predicated runoff with the downstream boundary condition time-series. The authors pointed out that because of the climate-driven changes, the model did not capture the spring flood dynamics. I may agree with the explanation, but we already know that complexity, and that is why we require a new robust model that can take care of the said complexity. 

4.    The inability of the model to predict the 2019 hydrograph was then corrected through the application of data assimilation – which means now the model is depended on the known runoff values to predict the next two days. The data assimilation tremendously improved the model efficiency from negative to above 0.90, indicating that the new known runoff is mainly correcting all the biases. I suggest the authors make an attempt to improve the predictivity of the core model. While the incorporation of the data assimilation is a reasonable technique here, it needs to be scientifically described with the adequate references.

5.    The figures need to be thoroughly revised. The numbers in the figures are difficult to read, and the X and Y axis captions need revision. It would be better if the authors can plot some of the figures rather than taking screenshots.

Specific comments:

The abstract needs to be thoroughly revised. There are lots of texts on the background of the work rather than the results. And I do not see the importance of giving the model link in the abstract.

Lin2 40-44: Better to delete.

Line 61-62: Overall, the paragraph needs revision. Notably, there is no need to bring the fake news argument.

Line 76-77: Needs revision.

Figure 1: Please use arrows to indicate the basins.

Section 2.2 is challenging to follow. It needs to be clearly written what data was used and from where it is obtained in a convenient way.

Line 110: What are those ICON NWP data?

Line 112: Resolution of 13X13 square grid?

Line 134-135: The sentence is tough to read - too much use of ‘which’.

Line 137-140: The splitting sample is good. However, please add a sentence to describe why particularly 1985 is the threshold year. Is there any change of regime after 1985? If we change the threshold year from 1985 to 1990, would it change the calibration and validation results?

Line 136: NSE is mostly used rather than NS.

Line 162: Please cite the Moriasi, 2007 paper where he classified the hydrological model performance based on NSE values. 

Please clearly indicate the temporal scale of the model output (daily?)

Figure 2: Use of “we” does not read well. You can write -Update of the open forecast website to demonstrate the new runoff forecast. Same is true for the other two instances where ‘we’ is used.

I think Table 1 and 2 can be combined – there is no need to provide two separate tables here.

Line 174: Replace “pilot sites” with “the”.

Line 208-210: The sentence needs to be better articulated. Currently, it reads like a personal communication style of writing. 

How the rating curve in Figure 3 is developed? The authors need to describe the rating curve development process elaborately.

Line 220: Same figures are referred to in the previous sentence

Line 340-342: Please relocate the sentences in the acknowledgment.

Reviewer 2 Report

I read the paper „OpenForecast: The first open source operational runoff forecasting system in Russia“ with a lot of interest, as I think that open source projects and collaborative undertakings are very important for transferring and increasing knowledge.

Below I include my comments and suggestions:

INTRODUCTION:

-          The introduction explains the motivation for the forceast product, there is however, no information about the aims and objectives of the paper. I think this should be added.

-          The last pargaraph of the introduction presents a summary of the results (Ln 78-81) and an outlook for the forecast product. I think this does not belong into the introduction.

-          I am a bit confused about the Ln 63-66, where it sasy that there is no public national operatial forecasting in Russia. What to you mean by “public”? In the next line there is information about the forcasts issued by Roshydromet on many webpages. Why is this not considered a public forcasing system?

MATERIALS AND METHODS:

-          As a main motivation of the forecast system is to have an open system that can be used anywhere, it would be interesting to add if the used datasets have global coverage and of they are freely available.

-          While the ERA-Interim dataset is commonly used, I had not heard about the ICON dataset. Is there a study looking at the quality of this data?

-           A question regarding the forecast system: is everything automated or is user input required?

-          I did not understand how the warm in states of the system are computed. What do you mean by “coupling the reanalysis data with an accumulated archive of ICON”?

-          In Figure 2 it says that the ERA-interim data is used for initializing model states. As far as I know, there is a time lag until the ERA-Intermin is made available, so how does this exactly work?

RESULTS:

-          In Ln 189 there is a sentence about winter flash flood, until now there was only a talk about spring floods. Does this paper focus on spring or winter floods or any flood? I think if would be really helpful to the readers if the types of flood encountered in the area would be explained in a previous step. Maybe in the introduction or site description.

-          Just a question: if you are interested in modelling floods, why is the model not calibrated for flood periods, but for the complete period with observations?

-          Figures 4 and 5 why is there are line at y=2x? There is not explanation for this in the text.

-          In Table 3 it is seen that there is little difference between the results ot different lag times. It would be nice to discuss this finding. Starting from Ln 237 the text gives two possible explanations for the bad performance in Basin Moskava. I found this part confusing. I think that one explanation is the 17-year time gap; another expalantion is that the calibration period might not be representative of the validation period. I think these are two different explanations, as the calibration and validation periods might have somehow different climate properties, without having any time gap in between.

-          The text in Figure 6 is too small.

-          I am not so convinced that the plots show there are only small differences between the ICON and ERA datasets. Maybe you could also present the values for the monthly/annual precipitation, and the correlation coefficient.

DISCUSSION:

-          As the main objective of the forecast product is developing a system that can be implemented in a wide range of locations it would be really helpful to include a short paragraph indicating what is required/needs to be done for implementing this system in other catchments. 

CONCLUSION:

-          As the main objective of a foracsting sytem is to “help” people to prepare for a damaging event, it is important that the tool is good and produces good quality results. I think that a forecasting system producing bad results can be really dangerous. That is why I think future work should focus much more on identifying the weaknesses of this tool and finding ways for addressing them rather than focusing on expanding its use.

Author Response

Round  2

Reviewer 1 Report

This is the second time I am reviewing the manuscript. The authors made some improvements to the primary document; however, I still think the changes are not substantial. The potential revision requires careful thinking and time to fix the issues; I do not feel the quick fixes that the authors did are adequate. Also, I had a tough time to read the response to reviewer document as line numbers are not cross-referenced, which is not desirable.

While making textual changes, the writing quality significantly declined.  A lot of intricate sentences were added – which in most cases made the writing very difficult to read. Further, several short sentences were combined using a comma, which, in general, is not a common practice. A thoughtful revision concerning the writing style is also expected. In multiple places, sentences are connected using "this way" – which is odd.

Although the major accomplishment of the manuscript is the development of an open source automated system for runoff forecasting, I have still some doubt about the reliability of the forecasts. In my previous review, I suggested to improve the predictability. However, it appears that this significant limitation is not adequately addressed – I do not think just listing the limitations do not resolve the issue as the biased predictions could mislead the users. Maybe it is essential to provide a warning in the website as well as in this current manuscript that the quantification of flood peaks is not correct and should not be used as a reference for any activities at this point. The manuscript could only be considered as a document that reports the system development rather than a fully developed system that can give reliable predictions. Before promoting the model to a broader audience through social media, it is imperative to improve the reliability of the predictions.

Line 3-5: The sentence needs revision. It feels like the sentence is incomplete because of the unusual use of the hyphen. With the current sentence, it reads like in our study, we presented openForcast. Further, in our study and we are coming together, which does not read well.

Line 8-10- Another example of a very complex sentence which can be written in a better way.  Further, it would be better it the authors replace the word significant.

Line 11-13- This is a kind of filler sentence in the abstract. This type of sentence is suitable for the conclusions. The author can add another sentence about the results and the implications of the work.

Line 19-22: Another hard to read sentence. Multiple looped logics in a single sentence.

Line 25: Replace "Viewed this way" by therefore.

Line 29-32: Insertion of not only but also made the sentence obscure.  It reads like several thoughts were merged into a single sentence using the comma. The writing should be coherent. Further, there should be no comma before but also.

Line 32: No forecast is perfect; therefore, it would be better to revise the sentence.

Line 40: countries where observational data is not always readily available – read better. 

Line 51-57: Another example of lengthy and complicated writing. This type of sentences needs to be revised carefully.

Line 51-54: The work did not advance the science of developing a new model to improve forecast; instead, the authors are presenting an open access system that can assimilate data from multiple sources to forecast runoff. Yet, the existing runoff model and meteorological forecast data are being criticized in this line as if the new system takes care of these issues.

Line 71:72: Do the authors have any reference to support the statement? In general, such a statement is avoided in scientific writing.

Line 88-89: Taking all---compelled -- please remove. It can be written as follows: In this study, we developed Russia's...

Line 102-103: Very non-specific objective.

Line 111: What is that expertise based on which the locations were selected. Please remove the based-on expertise logic and state the actual reasonings of choosing the study sites.

line 127: Please remove the extra space before the era-interim data set.

Line 129: I have seen this trend throughout the writing….a short sentence is connected through a comma. For example, the sentence can be written as follows:  1979 to date. The data is freely available.

Line 146-147: Please merge the sentence with the previous paragraph.

Line 155-162: Did not get it. I would appreciate getting the authors response here. The correlations are computed using the CAHEM data. How can the correlation indicate the reliability of the developed curve? The reported correlations could be misleading. It only suggests the importance of developing a rating curve. Further, in the right plot, the rating curve did not reliably follow the trend when the water level is high.

Figure 2: Please write water level, H (cm) in X-axis and Runoff, Q (m3/s) in Y-axis title.

Figure 3: If possible, please add a citation to refer the chart. Further, what is the meaning of the 0.9 and 0.1 in the figure?

Table 1: Please use a hyphen to indicate range; e.g., 0-3000.

Line 300-301: What are the NSE and Bias values when the entire simulation period is compared? Can the model capture those no- flood period other than the march to April period, as shown in the figure? I suggest adding another plot where the entire simulation range can be found (maybe in the supplemental section). Another thing – perhaps I missed the sentence – what parameter set is used for the 2019 simulations. What could be the potential parameter values that can represent both shape and magnitude of the spring 2019 hydrograph?

Line 351-353: Does it indicate the model is not suitable for the urban regions? The sentence questions the robustness of the model. As I suggested in my previous comment, these issues need to be solved to improve the model prediction.

Line 354: Please replace the word little.

Figure 8 (right): Should be observed Q and simulated Q.  Same issue in Figure 6.

Line 412-415: The writing is not good- please revise.

Line 428-429: Yes, this effort needs to be made to improve the system. Otherwise, in this present form, the system is essentially unreliable.

Reviewer 2 Report

I think the paper improved considerably and can be published as it is.

Author Response

Round  3

Reviewer 1 Report

Thank you for the response letter. I understand from the very beginning (from the 1st review) that the primary goal of the manuscript is to share the developed forecasting system, not the hydrological model. However, the reason I kept commenting on this issue is that the way the introduction is built, it appears the system also introduces a new hydrological model for better prediction of runoff. However, after three rounds of revisions, I think the issue is reasonably resolved. The utility of the operational runoff forecasting system needs to be the basis of the introduction rather than the limitations of the existing hydrological models.

About the writing style, the use of “this way” or “viewed this way” is grammatically correct – but the matter is we do not often see this kind of write-up in scientific papers. I understand that the manuscript is edited by a native writer; however, this is also important to follow the general practice. A well-written document always helps to get higher readership and eventually more citations. The problem with the current manuscript is the writing is unclear in some areas.

Note: For the line numbers, please refer to the version that is attached with the response letter

line 5: Please replace article by study.

Line 8-10: Still needs work. “although these proved” – what did you mean? Are you indicating model simulation? “they were inefficient” who are they?

Line 11: Missing article ‘the’ before revealed.

Line 28-31: Still very tough to read amid the revision. Please make the sentence simple. “they are far from perfect” – not clear what is the thing that you are referring as “they.” Further, you used which for a looped argument within the despite-based sentence structure.  This is very hard to follow.

Line 42-44: Which within which- very hard to follow this kind of looped argument.

Line 48: This ensures that - very unclear writing.

Line 87: Remove the sentence this choice has been….. You are already explaining the reason below. There is no need to repeat. Further, expertise of the authors is not a good way to write in a scientific manuscript. 

Line 133-134: Thank you for correcting the rating curve correlations. However, it would be better to report NSE/R2 instead of correlation because the model is fitted with regression; therefore, it would make more sense if you report R2.

Figure 2: Highlighted in red dots.

Figure 5,6,7: Small r, not the R should indicate correlations.

Line 363 in Conclusions: Please add a sentence that at this point, the model predictions should not be used as a reference because of the reported errors in runoff predictions.
